# Effectiveness of app-based cognitive behavioral therapy for insomnia on preventing major depressive disorder in youth with insomnia and subclinical depression: A randomized clinical trial

Si-Jing Chen[1,2,3], Jian-Yu Que[4,5], Ngan Yin Chan[1], Le Shi[4], Shirley Xin Li[6,7], Joey Wing Yan Chan[1], Weizhen Huang[4], Chris Xie Chen[1], Chi Ching Tsang[1], Yuen Lam Ho[1], Charles M. Morin[2,3], Ji-Hui Zhang[1,8,9]*, Lin Lu[4,10,11]*, Yun Kwok Wing[1,12]*

1 Li Chiu Kong Family Sleep Assessment Unit, Department of Psychiatry, Faculty of Medicine, The Chinese University of Hong Kong, Shatin, Hong Kong SAR, China, 2 École de psychologie, Université Laval, Québec City, Québec, Canada, 3 Centre d'étude des troubles du sommeil, Université Laval, Québec City, Québec, Canada, 4 Peking University Sixth Hospital, Peking University Institute of Mental Health, NHC Key Laboratory of Mental Health (Peking University), National Clinical Research Center for Mental Disorders (Peking University Sixth Hospital), Peking University, Beijing, China, 5 Xiamen Xianyue Hospital, Xianyue Hospital Affiliated with Xiamen Medical College, Fujian Psychiatric Center, Fujian Clinical Research Center for Mental Disorders, Xiamen, Fujian, China, 6 Department of Psychology, The University of Hong Kong, Shatin, Hong Kong SAR, China, 7 The State Key Laboratory of Brain and Cognitive Sciences, The University of Hong Kong, Shatin, Hong Kong SAR, China, 8 Center for Sleep and Circadian Medicine, The Affiliated Brain Hospital of Guangzhou Medical University, Guangzhou, Guangdong, China, 9 Key Laboratory of Neurogenetics and Channelopathies of Guangdong Province and The Ministry of Education of China, Guangzhou Medical University, Guangzhou, China, 10 National Institute on Drug Dependence and Beijing Key Laboratory of Drug Dependence, Peking University, Beijing, China, 11 Peking-Tsinghua Center for Life Sciences and International Data Group/McGovern Institute for Brain Research, Peking University, Beijing, China, 12 Faculty of Medicine, Li Ka Shing Institute of Health Sciences, The Chinese University of Hong Kong, Shatin, Hong Kong SAR, China

☯ These authors contributed equally to this work.
* jihui.zhang@cuhk.edu.hk (JHZ); linlu@bjmu.edu.cn (LL); ykwing@cuhk.edu.hk (YKW)

## Abstract

### Background

Increasing evidence suggests that insomnia plays an important role in the development of depression, supporting insomnia intervention as a promising approach to prevent depression in youth. This randomized controlled trial evaluated the effectiveness of app-based cognitive behavioral therapy for insomnia (CBT-I) in preventing future onset of major depressive disorder (MDD) in youth.

### Methods and findings

This was a randomized, assessor-blind, parallel group-controlled trial in Chinese youth (aged 15–25 years) with insomnia disorder and subclinical depressive symptoms. Participants were randomly assigned (1:1) to 6-week app-based CBT-I or 6-week app-based health education (HE) delivered through smartphones. Online assessments and telephone

**Data availability statement:** Data available on request. In order to meet ethical requirements for the use of confidential participant data, requests must be approved by the Li Chiu Kong Family Sleep Assessment Unit. Requests for data should be sent to sleepresearch@cuhk. edu.hk.

**Funding:** This study was funded by the National Natural Science Foundation of China (no. 81761128036 to LL) and The Chinese University of Hong Kong Postdoctoral Fellowship (to SJC). The funders had no role in study design, interpretation of the data or preparation of the manuscript.

**Competing interests:** I have read the journal's policy and the authors of this manuscript have the following competing interests: YKW received personal fees from Eisai Co, Ltd, for delivering a lecture and Consultation, and sponsorship from Lundbeck HK Ltd and Aculys Pharma, Inc outside the submitted work. CMM received research support from Idorsia, Eisai and Lallemand Health Solutions, personal fees from Idorsia, Eisai and Haleon, and received royalties from Mapi Research Trust outside the submitted work. JHZ provided consultancy for BestCare & SuMian BioTech Co, Ltd. JWYC received personal fee from Eisai Co., Ltd and travel support from Lundbeck HK limited for overseas conference outside the submitted work. The other authors have declared that no competing interests exist (SJC, QJY, NYC, LS, SXL, WZH, CXC, CCT, YLH, and LL).

**Abbreviations:** CBT-I, cognitive behavioral therapy for insomnia; GAD-7, Generalized Anxiety Disorder 7-item; HR, hazard ratio; HE, health education; ISI, Insomnia Severity Index; LOCF, last observation carried forward; MDD, major depressive disorder; MINI, Mini-International Neuropsychiatric Interview; PHQ-9, Patient Health Questionnaire-9; RCT, randomized controlled trial; RR, relative risk; SE, sleep efficiency; SOL, sleep onset latency; TIB, time in bed; TST, total sleep time; WASO, wake after sleep onset.

clinical interviews were conducted at baseline, post-intervention, 6- and 12-month follow-ups. The primary outcome was time to onset of MDD. The secondary outcomes included depressive symptoms and insomnia at both symptom and disorder levels. Between September 9, 2019, and November 25, 2022, 708 participants (407 females [57%]; mean age, 22.1 years [SD = 1.9]) were randomly allocated to app-based CBT-I group ($n$ = 354) or app-based HE group ($n$ = 354). Thirty-seven participants (10%) in the intervention group and 62 participants (18%) in the control group developed new-onset MDD throughout the 12-month follow-up, with a hazard ratio of 0.58 (95% confidence interval 0.38–0.87; $p$ = 0.008). The number needed to treat to prevent MDD at 1 year was 10.9 (6.8–26.6). The app-based CBT-I group has higher remission rates of insomnia disorder than the controls at post-intervention (52% versus 28%; relative risk 1.83 [1.49–2.24]; $p$ < 0.001) and throughout 12-month follow-up. In addition, the CBT-I group reported a greater decrease in depressive (adjusted difference –1.0 [–1.6 to –0.5]; Cohen's $d$ = 0.53; $p$ < 0.001) and insomnia symptoms (–2.0 [–2.7 to –1.3], $d$ = 0.78; $p$ < 0.001) than the controls at post-intervention and throughout 6-month follow-up. Insomnia was a mediator of intervention effects on depression. No adverse events related to the interventions were reported.

## Conclusions

App-based CBT-I is effective in preventing future onset of major depression and improving insomnia outcomes among youth with insomnia and subclinical depression. These findings highlight the importance of targeting insomnia to prevent the onset of MDD and emphasize the need for wider dissemination of digital CBT-I to promote sleep and mental health in the youth population.

## Trial registration

ClinicalTrials.Gov (NCT04069247).

## Author summary

### Why was this study done?

1. Emerging data on insomnia intervention, especially cognitive behavior therapy for insomnia (CBT-I), has shown promising results in alleviating both insomnia and depressive symptoms among patients with insomnia.

2. The existing research on insomnia intervention mostly focused on adult or older adult subjects. In addition, there are limited studies with adequate statistical power to demonstrate the preventive effect of CBT-I on depression at both symptom and disorder levels.

3. Little is known as to whether digital intervention of insomnia could potentially prevent youth depression.

### What did the researchers do and find?

1. This was a randomized controlled trial to evaluate the efficacy of a fully automated app-based CBT-I in preventing future onset of depression in 708 youth with insomnia disorder and subclinical depression.

2. We found that app-based CBT-I was effective in preventing future depression at both symptom and disorder levels.

3. Insomnia was a mediator of intervention effects on depression.

4. App-based CBT-I not only improved nocturnal symptoms of insomnia but also reduced daytime fatigue and led to a greater shift toward morningness.

## What do these findings mean?

1. The app-based CBT-I is a feasible and effective intervention for preventing future depression at both symptom and disorder levels in youth. Further studies should explore the integration of app-based CBT-I into clinical practice to enhance accessibility and prevent depression among the youth population.

2. Future pragmatic clinical trials are needed to explore the potential benefits of adapting digital sleep interventions in alleviating and preventing depression to advance the field toward personalized and stepped care approaches in community.

3. The main limitations of the current study include the inclusion of a mixture of individuals with and without prior depression, and a limited sample size of adolescents, albeit that there was an adequate overall sample size.

## 1. Introduction

Youth is a vulnerable transitional stage often linked to the emergence of sleep and mental health problems [1]. In particular, three-quarters of mental disorders emerge by the age of 24, with major depressive disorder (MDD) being one of the most common mental disorders among youth [2]. Youth depression tends to be persistent and recurrent, leading to various adverse long-term outcomes, including academic and social impairments, poor self-rated health, and increased suicidal risk [3,4]. Given the deleterious impact of youth depression, it is crucial to translate the modifiable risk factors for depression into an effective preventive strategy.

There is a growing recognition of insomnia as a significant and independent risk factor for depression in both adult and youth populations [5,6]. Interventions directly targeting insomnia, such as cognitive behavioral therapy for insomnia (CBT-I), may have the potential to prevent future onset of MDD. However, despite the effectiveness of CBT-I in reducing depressive symptoms in adult and youth populations [7–9], only a few studies with adequate statistical power examined the effects of CBT-I on preventing new-onset depression at disorder level, but with mixed findings [10]. For example, two prevention trials conducted in adults showed promising results of CBT-I in digital format for alleviating depressive symptoms among at-risk adults, whereas neither of these studies demonstrated the preventive effects of digital CBT-I for clinical depressive disorder [11,12]. On the other hand, a recent study showed that delivery of face-to-face CBT-I has significant benefits in preventing incident and recurrent MDD among older adults with insomnia [13]. Nevertheless, it remains unclear whether CBT-I (especially in digital format) can be used as an indicated prevention approach for at-risk individuals with subclinical depression [14], and whether its preventive effects can be achieved in youth, a vulnerable developmental period for the onset of mood problems.

In this study, we conducted a randomized controlled trial (RCT) to evaluate the efficacy of an app-based CBT-I on preventing future onset of MDD in youth with insomnia disorder and

subclinical depression. In addition, we evaluated the effects of app-based CBT-I on improving insomnia outcomes in secondary analyses. An app-based intervention was adopted in this study to address barriers in the implementation of CBT-I among youth, such as the low level of help-seeking and limited accessibility to healthcare system [15,16]. We hypothesized that (1) app-based CBT-I would reduce the incidence of MDD over a 12-month follow-up, (2) app-based CBT-I would reduce depressive and insomnia symptoms, as well as increase remission rates of insomnia disorder at post-intervention and follow-ups, and (3) the improvements in depressive symptoms would be mediated by changes in insomnia symptoms.

## 2. Methods

### 2.1. Study design and participants

This was a randomized, assessor-blind, parallel group-controlled trial conducted in Chinese youth (aged 15–25 years) with insomnia disorder and subclinical depression. Potential participants were recruited from universities, high schools, and communities in mainland China and Hong Kong from September 9, 2019 to November 25, 2022. They were invited to complete online screening assessments to confirm the presence of moderate-to-severe insomnia symptoms (Insomnia Severity Index [ISI] [17] score ≥15) and subclinical depressive symptoms (Patient Health Questionnaire-9 [PHQ-9] [18] score greater than 4 but less than 20). Participants who met the screening criteria were invited for a telephone diagnostic interview to ascertain the following inclusion criteria: (1) presence of insomnia disorder based on International Classification of Diseases, 10th Revision diagnostic criteria, defined as insomnia symptoms ≥3 times per week, accompanied by significant distress or functional impairments for at least 1 month, which cannot be adequately explained by another sleep–wake disorder, mental disorder, or medical condition [19] and (2) absence of a current diagnosis of MDD or a prior episode of MDD within past 2 months by the Mini-International Neuropsychiatric Interview (MINI) [8]. Additional details about inclusion and exclusion criteria can be found in the research protocol (S1 Protocol).

All eligible participants were randomly assigned to 6-week app-based CBT-I or 6-week app-based health education (HE) at 1:1 ratio, irrespective of any pharmacological treatments for insomnia that the participant has been receiving (Fig 1). Assessments, including telephone clinical interviews and online self-administered questionnaires, were conducted at baseline, post-intervention, 6- and 12-month follow-ups after intervention. Two additional assessments were conducted upon the completion of sessions 2 (post-session 2) and 4 (post-session 4) to evaluate sleep and mood symptoms. If participants experienced a depressive episode before the 12-month follow-up, they were considered as completed cases that met the endpoint of the study. The study was reported in accordance with the CONSORT Guidelines (S1 Checklist).

Written informed consent was obtained from participants aged 18 and above. For participants under 18, assent from them and consent from their parent(s)/caregiver(s) were obtained. Ethical approval for the study was granted by Joint Chinese University of Hong Kong-New Territories East Cluster Clinical Research Ethics Committee (CREC Ref. No.: 2019.044) and Medical Ethics Committee of Peking University Sixth Hospital (Issue No. 21 [2019]). The trial is registered with Clinical Trials.Gov (NCT04069247).

### 2.2. Randomization and masking

Randomization was stratified by sex and insomnia severity (15 ≤ ISI ≤ 21 versus ISI > 21) at baseline. A research assistant who was not involved in the study generated the random number sequence for this trial using RAND and RANK functions in Excel with a block size of four. Study participants were not informed of their group assignment, although it was likely that

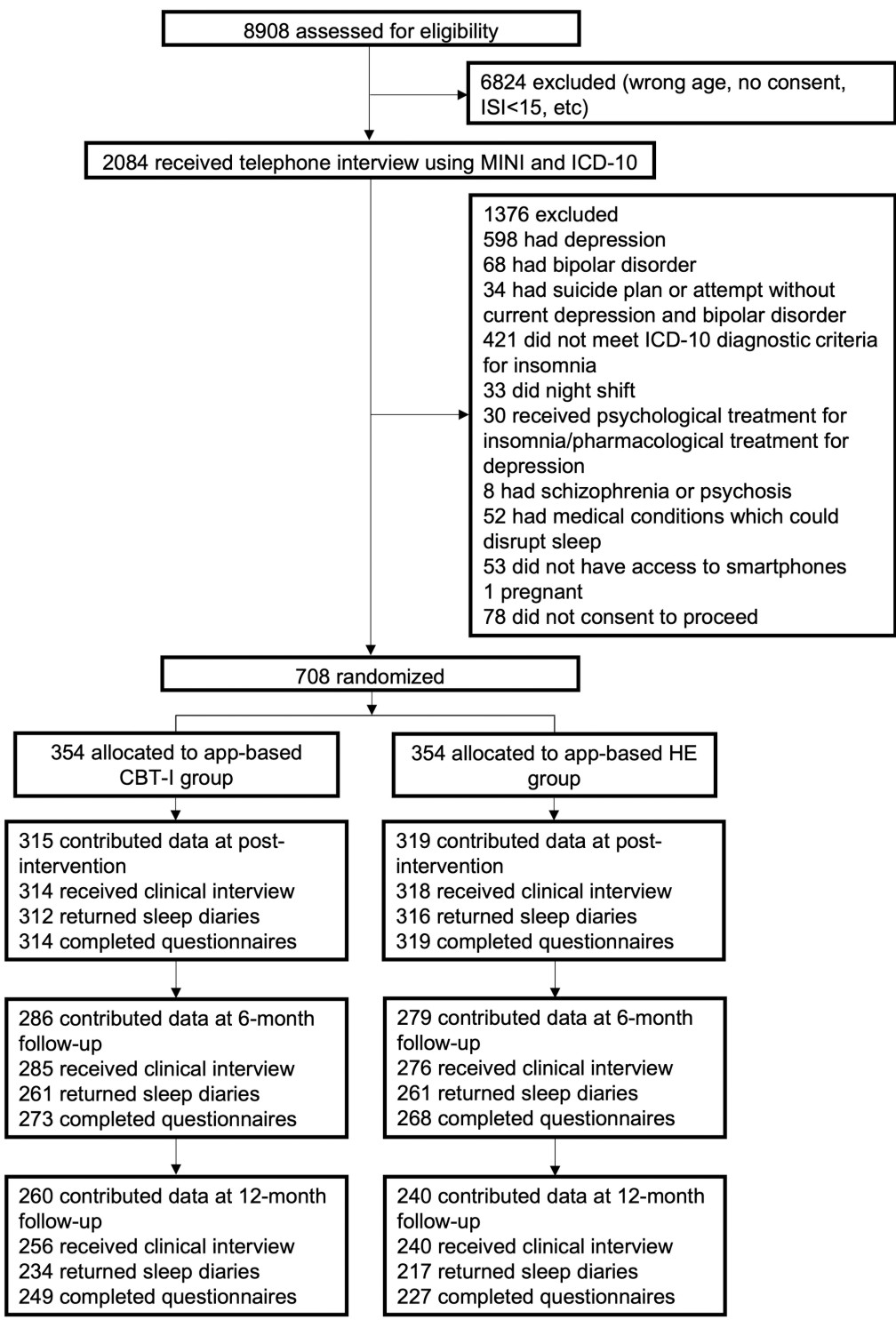

**Fig 1. Participant flowchart.** CBT-I, cognitive behavioral therapy for insomnia; HE, health education; ISI, Insomnia Severity Index; PHQ-9, Patient Health Questionnaire-9; MINI, Mini-International Neuropsychiatric Interview; ICD-10, International Classification of Diseases, 10th Revision.

most of the participants were aware of their allocation based on the content of the program. Trained investigators who conducted telephone clinical interviews were masked to group allocation.

### 2.3. Intervention

**2.3.1. Experimental condition.** The app-based CBT-I intervention is a digital, self-paced, and interactive insomnia intervention program delivered through smartphones (esleep: Android: https://sd-oss-cdn.sumian.com/apk/eSleep_1.0.3-release.apk; iOS: Apple App store). It was initially developed by our research team (Li Chiu Kong Family Sleep Assessment Unit, Department of Psychiatry, The Chinese University of Hong Kong, Hong Kong SAR) and was modified and issued by BestCare & SuMian BioTech Co., Ltd. The program was designed to adapt for the language background of the participants in China. Based on a well-established CBT-I treatment protocol, the program consists of six sequential modules, including (1) overview of sleep, (2) sleep restriction, (3) stimulus control, (4) cognitive restructuring (targeting sleep-related dysfunctional cognitions), (5) structured worry time, and (6) relapse prevention (Table A in S1 Appendix). Each module consisted of 20–30 min of animated videos and was automatically unlocked weekly. During sleep restriction module (session 2), participants would obtain a "sleep prescription" from the application, specifying their sleep window for the next 7 days. The setting of the sleep window was based on each individual's expected wake-up time and average total sleep time (TST) from the previous week, while adhering to the 5-h minimum rule. The sleep prescription was adjusted weekly based on each individual's sleep efficiency (S1 Protocol). Participants were allowed to complete the intervention in a 12-week window. Individualized text messages based on participants' course progress were sent to the participants regularly to remind them of timely completion of the treatment modules (once per week).

**2.3.2. Attention-matched control.** The control (app-based HE) condition is a health promotion program developed to control for the potential dosing effect of attention and non-specific components (e.g., expectation and contact hours). The general sleep knowledge session was purposively added to the program to meet the expectations of the participants, while it did not include any active insomnia and depression therapeutic components (Table A in S1 Appendix). Similarly, participants in the control group were contacted weekly to remind them to complete the modules.

### 2.4. Outcome measures

The primary outcome was time to onset of MDD determined by telephone clinical interviews using the Chinese version of MINI at post-intervention, 6- and 12-month follow-ups. The MINI has been validated in Chinese psychiatric patients with adequate psychometric properties (sensitivity for diagnosing depression = 92.2%; specificity = 86.0%) [20]. During each interview, the evaluation covered the period from the previous assessment to establish the onset of a depressive episode. In cases at which the participants were unable to recall the exact date, they were asked to indicate the closest week or month, and the midpoint of that week or month was used as the reference point. To assess interrater reliability, interviews were recorded and independently reviewed by a second investigator. The κ coefficient for interrater agreement regarding the diagnosis of MDD was 0.82 based on randomly selected data from 12% (85/708) of the participants, indicating excellent agreement.

The secondary outcomes included depressive symptoms and the measures of insomnia at both symptom and disorder levels. Remission of insomnia disorder was determined by the following criteria [21]: (1) no functional impairments or distress with sleep, (2) insomnia

symptoms less than three times per week for at least 1 month, and (3) sleep-promoting medication use less than once per week in the past month. Although the use of sleep-promoting medication is not a standard criterion for diagnosing insomnia disorder, it may mask the underlying symptoms and was thus taken into account in the current study [21]. Changes in depressive and insomnia symptoms were assessed with PHQ-9 and ISI, respectively.

Other prespecified outcomes included anxiety symptoms (Generalized Anxiety Disorder 7-item [GAD-7]) [22], suicidal ideation (Scale for Suicide Ideation-Current) [23], Fatigue (Multi-dimensional Fatigue Inventory) [24], circadian preference (reduced Morningness-Eveningness Questionnaire) [25], sleep-related beliefs and attitudes (Brief version of Dysfunctional Beliefs and Attitudes about Sleep) [26], and sleep parameters including time in bed (TIB), TST, sleep onset latency (SOL), wake after sleep onset (WASO), sleep efficiency (SE) measured by 7-Day Sleep Diary.

Adverse events related to the interventions that occurred from randomization until the 12-month follow-up were recorded during telephone interviews by asking participants if they experienced any adverse events attributed to the interventions from the previous assessment to the current interview, and to specify if they answered yes.

In the study protocol, multiple outcomes were designated as primary outcomes based on the study hypothesis, including depression and insomnia at both symptom and disorder levels. The presence of multiple primary outcomes and the lack of a defined hierarchy may pose some concern to the interpretation of the major findings of the study. Therefore, we reorganized the hierarchy of outcomes when reporting the study results to ensure that the trial was chiefly judged by the outcome for which it was fully powered, namely the incidence of depressive disorder.

## 2.5. Sample size calculation

The sample size calculation was based on the incidence of depressive disorder. As the study sample included a mixture of individuals with and without prior history of MDD, a 50% increase in the 1-year incidence of depression was expected compared to that (11%) reported in the previous meta-analysis on patients with insomnia [6]. According to previous research on self-help intervention for prevention of MDD [27] and data on CBT intervention for depression in youth population [28–30], we expected a 50% reduction in the incidence rate for the intervention group. Based on a 1-year incidence rate of 16.5% (11% × 1.5), power analysis indicated that 282 participants per condition were needed to detect a hazard ratio (HR) of 0.5 under 1:1 randomization at $\alpha < 0.05$ (two-tailed) with 80% statistical power. To account for 40% potential attrition rate, a total of 940 participants would be initially needed. In the amendment of study protocol, the recruitment was reduced to a sample size of 708 (354 per condition) based on an estimated attrition rate of approximately 20%, as the attrition rate was much lower than initially expected.

## 2.6. Statistical analysis

The statistical analysis plan was finalized before the completion of data collection (S1 Protocol), except for subgroup analyses of the primary outcome and exploratory analysis controlling for potential confounders affecting the primary outcome, including prior history of MDD, educational level, family income, comorbid medical illnesses, and use of sleep-promoting medications [31], to test the robustness of the results. Analyses were conducted based on the intent-to-treat approach. Kaplan-Meier survival analysis and Cox proportional hazard regression model were employed to determine differences in time to onset of MDD between the intervention and control groups over 12-month follow-up. Remission rates of insomnia were

analyzed using a weighted generalized estimating equations model. Weights were computed as the product of a missing model weight to attenuate the effect of attrition conditional on strata variables and group allocation [32]. The intervention effects on continuous variables were analyzed using linear mixed-effects model analysis with residual maximum likelihood estimation. In the sensitivity analyses, missing scores due to participants reaching the study endpoint (experiencing a depressive episode) before the 12-month follow-up were handled using last observation carried forward (LOCF) method, as LOCF likely to be conservative when the target condition is expected to improve spontaneously over time (e.g., insomnia and depression) [33]. Mediation analyses using maximum likelihood estimation within a structural equation modeling framework were used to investigate the mediating effects of insomnia on depressive symptoms (the mediating effects were tested when the efficacy analysis showed significant differences between the two groups in depressive symptoms). Strata variables at baseline were included as covariates in all analyses. Data analyses were performed using the Stata 17.0 statistical software with standard two-tailed $\alpha$ of 0.05.

## 3. Results

### 3.1. Participants

A total of 708 youth (407 females [57%]; mean [SD] age, 22.1 [1.9] years) with insomnia disorder (mean duration, 3.0 years) were allocated to either app-based CBT-I ($n$ = 354) or app-based HE group ($n$ = 354) (Table 1). Majority of the participants (90%; 639/708) were undergraduate students or university educated. Nearly half of the participants (48%; 339/708) had a history of MDD, and 17% (123/708) of them had suicidal ideation in the past month as confirmed by MINI interview, while only few of them had taken antidepressants before (5%; 38/708). In addition, 140 (20%) participants took sleep-promoting medications at baseline. The percentage of participants using sleep-promoting medications decreased in both groups at post-intervention (app-based CBT-I versus app-based HE, 12% versus 15%) and follow-up assessments (12-month, 10% versus 14%) with no significant group differences (Fig A in S1 Appendix).

Of the included participants ($n$ = 708), 634 (90%) contributed data to the post-intervention assessment, 565 (80%) completed the 6-month assessments, and 500 (71%) completed the assessments at 12 months (Fig 1). Attrition rates from assessments at post-intervention and follow-ups did not differ significantly between the two groups. In the app-based CBT-I group, 333 participants (94%) logged on at least one session and 296 (84%) completed all six sessions (Table A in S1 Appendix). As for the app-based HE group, 332 (94%) participants logged on at least one session and 301 (85%) completed six sessions. No adverse events related to the interventions were reported.

### 3.2. Preventive effect of app-based CBT-I on major depression

Thirty-seven participants (incidence proportion, 10%; 37/354) in the intervention group and 62 participants (incidence proportion, 18%; 62/354) in the control group developed new-onset MDD during the 12-month follow-up. Fig 2 shows the survival curves for the app-based CBT-I and control groups. The estimates of the cumulative 1-year incidence rate (person-time rate) of MDD were 12% (95% confidence interval [CI] 9% to 17%) for the app-based CBT-I group and 21% (17% to 27%) for the HE group. The number needed to treat to prevent MDD at 1 year was 10.9 (6.8–26.6). Cox regression revealed an HR of 0.58 (0.38–0.87; $p$ = 0.008) for MDD favoring app-based CBT-I intervention.

After additionally adjusting for prior history of MDD, educational level, family income, comorbid medical illnesses, and use of sleep-promoting medications in exploratory analysis,

**Table 1. Baseline characteristics.**

| | Total sample (*n* = 708) | App-based CBT-I (*n* = 354) | App-based HE (*n* = 354) |
|---|---|---|---|
| Age, years | 22.1 (1.9) | 21.9 (2.0) | 22.2 (1.9) |
| <18y | 10 (1.4) | 5 (1.4) | 5 (1.4) |
| ≥18y | 698 (98.6) | 349 (98.6) | 349 (98.6) |
| Sex | | | |
| Female | 407 (57%) | 202 (57%) | 205 (58%) |
| Male | 301 (42%) | 152 (43%) | 149 (42%) |
| Relationship status | | | |
| De facto | 16 (2%) | 11 (3%) | 5 (1%) |
| Married | 2 (<1%) | 2 (<1%) | 0 (0%) |
| Never married | 690 (97%) | 341 (96%) | 349 (99%) |
| Site | | | |
| Hong Kong | 41 (6%) | 18 (5%) | 23 (6%) |
| Mainland China | 667 (94%) | 336 (95%) | 331 (94%) |
| Education level | | | |
| Undergraduate student and above | 639 (90%) | 318 (90%) | 321 (91%) |
| Other | 69 (10%) | 36 (10%) | 33 (9%) |
| Full-time student | 545 (77%) | 273 (77%) | 272 (77%) |
| Family monthly income | | | |
| CNY < 8,000 | 484 (68%) | 251 (71%) | 233 (66%) |
| CNY ≥ 8,000 | 209 (30%) | 100 (28%) | 109 (31%) |
| Lifestyle | | | |
| Tea consumption | | | |
| <3/week | 600 (85%) | 312 (88%) | 288 (81%) |
| ≥3/week | 93 (13%) | 39 (11%) | 54 (15%) |
| Coffee consumption | | | |
| <3/week | 608 (86%) | 312 (88%) | 296 (84%) |
| ≥3/week | 85 (12%) | 39 (11%) | 46 (13%) |
| Energy drink consumption | | | |
| <3/week | 687 (97%) | 349 (99%) | 338 (95%) |
| ≥3/week | 6 (<1%) | 2 (<1%) | 4 (1%) |
| Beverage consumption | | | |
| <3/week | 615 (87%) | 310 (88%) | 305 (86%) |
| ≥3/week | 78 (11%) | 41 (12%) | 37 (10%) |
| Alcohol consumption | | | |
| <3/week | 683 (96%) | 346 (98%) | 337 (95%) |
| ≥3/week | 10 (1%) | 5 (1%) | 5 (1%) |
| Smoking | | | |
| <3/week | 653 (92%) | 329 (93%) | 324 (92%) |
| ≥3/week | 40 (6%) | 22 (6%) | 18 (5%) |
| Depression | | | |
| History of MDD[a] | 339 (48%) | 163 (46%) | 176 (50%) |
| Prior use of antidepressants[b] | 38 (5%) | 16 (5%) | 22 (6%) |
| Insomnia | | | |
| Use of sleep-promoting medications at baseline[c] | 140 (20%) | 76 (21%) | 64 (18%) |
| Duration of insomnia, years | 3.0 (2.7) | 3.0 (2.7) | 3.0 (2.7) |

*(Continued)*

**Table 1.** (Continued)

| | Total sample (*n* = 708) | App-based CBT-I (*n* = 354) | App-based HE (*n* = 354) |
|---|---|---|---|
| Suicidal ideation at baseline | 123 (17%) | 58 (16%) | 65 (18%) |
| Diagnosed medical illnesses[d] | 152 (21%) | 86 (24%) | 66 (19%) |

Abbreviations: CBT-I, cognitive behavioral therapy for insomnia; HE, health education; MDD, major depressive disorder.

[a]History of MDD was ascertained by the Mini-International Neuropsychiatric Interview (MINI).

[b]Antidepressants included selective serotonin reuptake inhibitors (SSRIs), serotonin antagonist and reuptake inhibitors (SARIs), serotonin and norepinephrine reuptake inhibitors (SNRIs), noradrenaline and specific serotonergic antidepressants (NaSSAs), tricyclic antidepressants (TCAs), and agomelatine.

[c]Sleep-promoting medications included benzodiazepines, non-benzodiazepines, trazodone (25–50 mg), traditional Chinese medicine, sedating antihistamines, and melatonin.

[d]Medical illnesses included eye disease, arthritis, heart disease, diabetes, renal disease, gastro-esophageal reflux disease, hypertension, epilepsy, chronic lung disease, and chronic pain.

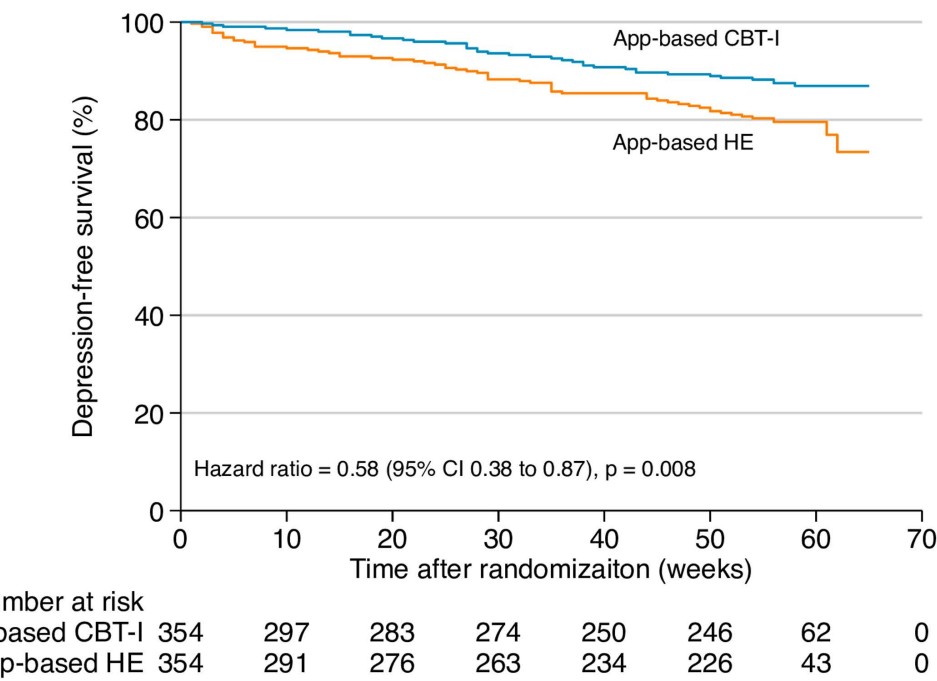

**Fig 2. Time to onset of major depressive disorder by intervention group.** CBT-I, cognitive behavioral therapy for insomnia; HE, health education.

the risk of developing MDD remained significantly lower for the app-based CBT-I intervention group compared with the controls (adjusted HR 0.60 [0.40–0.90]; *p* = 0.01) (Table B in S1 Appendix). Further analyses demonstrated that the preventive effect of app-based CBT-I on MDD across various subgroups was generally consistent with the effect observed in the overall sample (Fig B in S1 Appendix), and there was no interaction between the intervention effect and any of the variables that defined the subgroups (*p*s > 0.05 for all interactions). In the sensitivity analyses conducted among participants with persistent insomnia (insomnia duration ≥ 3 months, *n* = 616), all results remained consistent (Table C and Fig C in S1 Appendix).

### 3.3. Intervention effect of app-based CBT-I on secondary outcomes

**3.3.1. Remission rates of insomnia disorder.** Fig 3 shows remission rates of insomnia disorder (adjusted means and standard errors) by groups and assessments. At post-intervention, the weighted percentage of remitters based on clinical interviews in app-based CBT-I group was higher than that of the app-based HE group (52% versus 28%; relative risk [RR] 1.83 [95% CI 1.49–2.24]; $p < 0.001$) after controlling for strata variables and accounting for the missing data patterns. Improvements achieved in the app-based CBT-I group were well sustained throughout the follow-up period, with higher remission rates than the controls at both 6-month (56% versus 44%; RR 1.27 [1.08–1.48]; $p = 0.003$) and 12-month follow-ups (57% versus 48%; RR 1.17 [1.01–1.35]; $p = 0.03$). The results remained consistent in the sensitivity analysis (Fig D in S1 Appendix).

**3.3.2. Depressive and insomnia severity.** Results from the mixed-effects model revealed significant group by time interactions for both ISI and PHQ-9 scores (both $ps < 0.001$; Table 2 and Fig E in S1 Appendix). Compared with the control condition, the app-based CBT-I intervention led to greater improvements in both depressive and insomnia symptoms at post-session 4 (depressive symptoms, adjusted difference −1.0 [1.6 to −0.4]; Cohen's $d = 0.50$; $p = 0.002$; insomnia symptoms, −1.7 [95% CI −2.4 to −1.0]; $d = 0.66$; $p < 0.001$) and post-intervention (depressive symptoms, −1.0 [−1.6 to −0.5]; $d = 0.53$; $p < 0.001$; insomnia symptoms, −2.0 [−2.7 to −1.3]; $d = 0.78$; $p < 0.001$). The results for depressive symptoms remained the same after excluding sleep item from the PHQ-9.

The intervention effects of app-based CBT-I on insomnia symptoms were well sustained throughout the follow-up period (6-month, $d = 0.45$; $p = 0.002$; 12-month, $d = 0.32$; $p = 0.03$). While significant group differences in depressive symptoms were only sustained at 6-month follow-up ($d = 0.42$; $p = 0.009$), albeit depressive symptoms remained in the mild range for

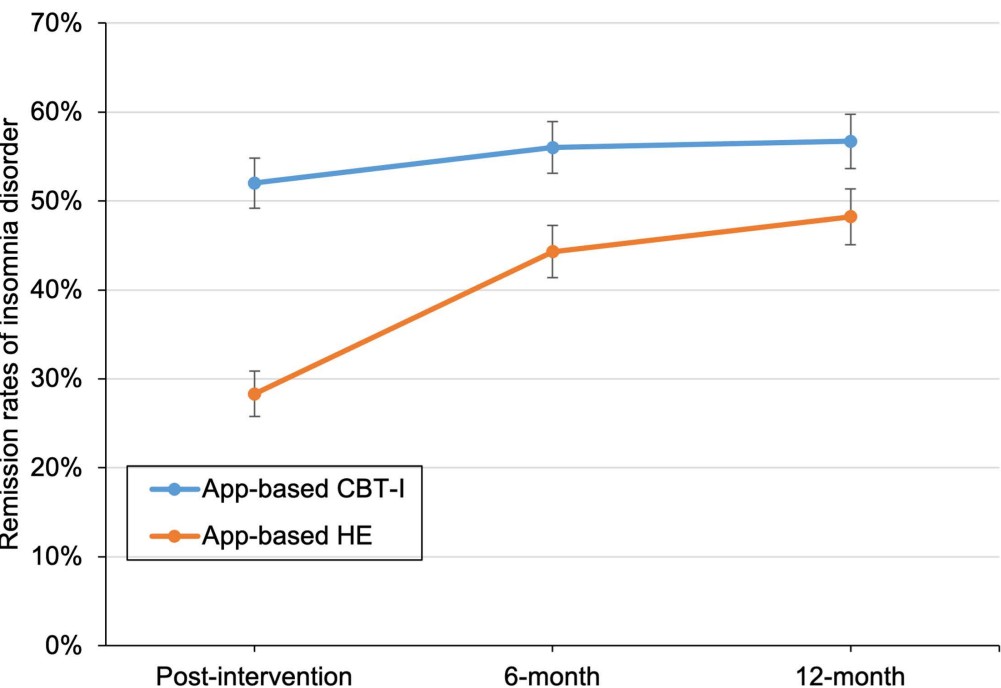

**Fig 3. Remission rates of insomnia disorder by intervention group.** Error bars indicate standard errors. CBT-I, cognitive behavioral therapy for insomnia; HE, health education.

**Table 2. Secondary outcome results.**

| | App-based CBT-I, mean (SE)[a] | App-based HE, mean (SE) | Adjusted difference (95% CI) | *p* value | Cohen's *d*[b] |
|---|---|---|---|---|---|
| *PHQ-9* | | | | | |
| Baseline | 12.1 (0.2); *n* = 354 | 12.1 (0.2); *n* = 354 | | | |
| Post-session 2 | 7.6 (0.2); *n* = 291 | 7.7 (0.2); *n* = 285 | −0.1 (−0.7, 0.5) | 0.72 | 0.06 |
| Post-session 4 | 6.1 (0.2); *n* = 256 | 7.0 (0.2); *n* = 264 | −1.0 (−1.6, −0.4) | 0.002 | 0.50 |
| Post-intervention | 5.1 (0.2); *n* = 314 | 6.2 (0.2); *n* = 319 | −1.0 (−1.6, −0.5) | <0.001 | 0.53 |
| 6 months | 5.4 (0.2); *n* = 273 | 6.3 (0.2); *n* = 268 | −0.8 (−1.4, −0.2) | 0.009 | 0.42 |
| 12 months | 5.5 (0.2); *n* = 249 | 6.0 (0.2); *n* = 227 | −0.5 (−1.1, 0.2) | 0.14 | 0.24 |
| *PHQ-8*[c] | | | | | |
| Baseline | 9.6 (0.2); *n* = 354 | 9.6 (0.2); *n* = 354 | | | |
| Post-session 2 | 6.1 (0.2); *n* = 291 | 6.2 (0.2); *n* = 285 | −0.1 (−0.6, 0.5) | 0.83 | 0.03 |
| Post-session 4 | 4.9 (0.2); *n* = 256 | 5.6 (0.2); *n* = 264 | −0.7 (−1.2, −0.1) | 0.02 | 0.39 |
| Post-intervention | 4.1 (0.2); *n* = 314 | 5.0 (0.2); *n* = 319 | −0.9 (−1.4, −0.4) | 0.001 | 0.51 |
| 6 months | 4.5 (0.2); *n* = 273 | 5.1 (0.2); *n* = 268 | −0.7 (−1.2, −0.1) | 0.01 | 0.39 |
| 12 months | 4.5 (0.2); *n* = 249 | 4.9 (0.2); *n* = 227 | −0.4 (−1.0, 0.2) | 0.18 | 0.23 |
| *ISI* | | | | | |
| Baseline | 18.3 (0.2); n = 354 | 18.3 (0.2); *n* = 354 | | | |
| Post-session 2 | 11.5 (0.2); n = 291 | 12.1 (0.2); *n* = 285 | −0.6 (−1.3, 0.1) | 0.12 | 0.22 |
| Post-session 4 | 9.0 (0.2); n = 256 | 10.7 (0.2); *n* = 264 | −1.7 (−2.4, −1.0) | <0.001 | 0.66 |
| Post-intervention | 7.3 (0.2); n = 313 | 9.3 (0.2); *n* = 319 | −2.0 (−2.7, −1.3) | <0.001 | 0.78 |
| 6 months | 7.2 (0.2); n = 272 | 8.4 (0.2); *n* = 268 | −1.1 (−1.9, −0.4) | 0.002 | 0.45 |
| 12 months | 7.3 (0.2); n = 249 | 8.2 (0.2); *n* = 227 | −0.8 (−1.6, −0.1) | 0.03 | 0.32 |

Abbreviations: CBT-I, cognitive behavioral therapy for insomnia; HE, health education; ISI, Insomnia Severity Index; PHQ, Patient Health Questionnaire.

[a]All values are adjusted for sex and insomnia severity at baseline.

[b]Cohen's *d* is defined as the adjusted intervention effect divided by the square root of estimated sample variance derived from the linear mixed-effects model.

[c]Excluding sleep item.

the intervention group at 12-month follow-up. In the sensitivity analyses with LOCF, greater improvements in both depressive and insomnia symptoms in the app-based CBT-I group compared to the control group were consistently observed throughout post-session 4–12-month follow-up (Table D in S1 Appendix).

In the mediation analyses, insomnia symptoms at post-session 4 accounted for 74% of the intervention effect on depressive symptoms measured by PHQ-8 (excluding sleep item) at post-intervention (Table E in S1 Appendix). Similarly, insomnia symptoms at post-intervention explained 78% of the improvements in depressive symptoms at 6-month. There were no significant residual direct effects observed in both models, indicating that the impacts of the intervention on depressive symptoms depend on its effects on insomnia. While in the reverse mediation analysis at which depressive symptoms at post-session 4 and post-intervention were set as the mediators, direct effects of the intervention on insomnia symptoms were noticed (81% at post-intervention and 57% at 6-month), suggesting that the intervention improved insomnia symptoms independent of its impacts on depressive symptoms.

### 3.4. Intervention effect of app-based CBT-I on other outcomes

The app-based CBT-I intervention produced greater improvements than the app-based HE for most of the outcomes at post-intervention (Tables F–G and Fig F in S1 Appendix), including SOL, TIB, SE, anxiety symptoms, fatigue, dysfunctional sleep beliefs, and a greater shift

toward morningness, which were maintained at least up to the 6-month follow-up, except for anxiety symptoms. There were no significant differences between the two groups in WASO, TST, or suicidal ideation.

## 4. Discussion

This large-scale RCT study showed that app-based CBT-I is effective in reducing future onset of MDD and increasing the remission rates of insomnia in youth with insomnia disorder and subclinical depression. The observed causal relationship between changes in insomnia symptoms and improvements in depressive symptoms lends additional support that insomnia plays an important role in precipitating the onset of depression. In addition, timely intervention of insomnia disorder in youth not only improved nocturnal symptoms but also reduced daytime fatigue and led to a greater shift toward morningness. Overall, our findings expand upon the results of recent sleep interventions in older adults [10], supporting the efficacy of app-based CBT-I in preventing depression and improving insomnia in youth population.

To the best of our knowledge, our finding on the preventive effect of app-based CBT-I on MDD in youth population is novel. In addition, self-help app-based CBT-I showed a similar magnitude of preventive effect in youth (HR 0.58 [95% CI 0.38–0.87]) when compared to previous prevention trials that employed face-to-face CBT-I in elderly (HR 0.51 [0.29–0.88]) [13] and web-based psychological intervention in middle-aged adults (HR 0.59 [0.42–0.82]) [27]. Furthermore, the preventive effect of app-based CBT-I on MDD was generally consistent across most subgroups, even in the presence of baseline suicidal ideation and a prior history of MDD. Given that the intervention program did not incorporate any therapeutic components specifically aimed at directly treating depression, the mechanisms underlying the preventive effect of CBT-I on MDD warrant further investigation. This effect may be partially attributed to the ability of CBT-I to indirectly address depression through sleep improvements, as indicated by the mediation analyses, or directly by targeting shared mechanisms between insomnia and depression, such as hyperarousal, ruminations, and worries [34,35].

As for the intervention effects of app-based CBT-I on depressive symptom severity, the between-group effect size at post-intervention ($d = 0.53$) is comparable to that reported in a recent meta-analysis on digital CBT-I ($d = 0.42$ [0.28–0.56]) [9], which was sustained at 6-month follow-up. Although depressive symptoms of the intervention group remained in the mild range at 12-month follow-up, we did not detect significant between-group effects due to the large within-group changes in the control arm. This could be partly due to the sleep improvements observed in the control group, which may in turn lead to changes in mood. Another reason for the lack of differentiation in symptom improvements at 12 months may be that some participants reached the study endpoint (experiencing a depressive episode) before the final follow-up. These participants were removed from subsequent assessments and provided with information for further treatment due to ethical considerations. Further sensitivity analyses with LOCF demonstrated greater improvements in depressive symptoms in the app-based CBT-I group at 12-month follow-up.

For insomnia outcomes, the large between-group effect size on insomnia severity at post-treatment in the current youth population ($d = 0.78$) is consistent with the findings of the previous meta-analysis on digital CBT-I in adults ($d = 0.76$) [9]. The intervention effects on insomnia were sustained throughout the follow-up period, albeit effect size was attenuated ($d = 0.32$ at 12-month follow-up). This decline in intervention effects was also documented by a previous meta-analysis [36], indicating that additional booster sessions might be needed to maintain long-term effects. Furthermore, the current study did not combine app-based CBT-I with circadian interventions (e.g., light therapy and circadian rhythm support), which have shown potential benefits in managing sleep and depressive symptoms in both adolescent

and adult subjects [37,38]. Given the overlap between insomnia and circadian problems in youth, future digital sleep intervention may need further adaption by incorporating treatment components typically targeting circadian problems to optimize the treatment effects in youth population.

In contrast to the long-term effects of app-based CBT-I on insomnia, its anxiolytic effect observed at post-intervention did not sustain at follow-ups, despite the close relationship between insomnia and anxiety [39]. This finding may be attributed to the fact that our youth subjects had only mild anxiety symptoms at baseline, leaving little room for further improvement. Nevertheless, the mean GAD-7 score for youth receiving the app-based CBT-I intervention declined from mild at baseline to a range of near normal throughout the follow-up period.

Our findings have significant implications for clinical practice and research. First, our study supports the notion of targeting modifiable risk factors, especially insomnia, to prevent future depression in youth. Second, fully automated digital CBT-I has the potential to address unmet clinical needs, enhance accessibility, and encourage help-seeking behavior among youth. Further studies should consider integrating digital CBT-I into clinical practice, especially among primary care. Third, previous RCT studies that evaluated the efficacy of digital CBT-I in youth with insomnia were all conducted in Western culture, and the majority of them had modest sample sizes [40–42]. Our study further confirmed that digital CBT-I is effective for treating insomnia and preventing depressive symptoms and disorders in non-Western youth population, with substantial long-term effects. Last, previous RCTs conducted in adults have reported high attrition rates (about 50% at post-intervention), which remains as a major barrier to the wider implementation of digital CBT-I [11,12,42]. Participants in the current study with weekly individualized text reminder based on their course progress have demonstrated a relatively low attrition rate (10% at post-intervention; 74/708). This finding indicates the effectiveness of timely reminders in reducing attrition and suggests that digital treatment may be preferable to youth population who have been described as digital native. Furthermore, a similarly low attrition rate was also observed in a previous RCT on digital CBT-I conducted in non-Western culture [43], suggesting possible cross-cultural differences in terms of the completion of intervention activities.

The current study has several strengths. First, as a large-scale prevention trial, we had sufficient statistical power to demonstrate the preventive effect of CBT-I on depression. Second, the depression and insomnia outcomes in the current study were confirmed by both diagnostic interview and self-report questionnaires, which allow us to capture the multi-faceted nature of individual's sleep and mood at both symptom and disorder levels. Third, there was a high attendance rate for the intervention program in the current study, with 84% (597/708) of the participants completing all six sessions of the intervention, which was higher than that of most online CBT-I programs. However, several limitations should be noted when interpreting the findings. First, participants in our study included a mix of individuals with and without a prior history of MDD, with 48% (339/708) of participants reporting a history of prior MDD. Nonetheless, only 5% (38/708) of them received treatment with antidepressants, and all of them had discontinued antidepressant use at least 2 months before the intervention, except for low-dose trazodone (25–50 mg) as sleep aids. In addition, the lack of interaction between the intervention effect and prior depression suggests that there is no difference in response to app-based CBT-I between participants with and without prior depression. Nevertheless, future randomization studies stratified by a prior history of depression are needed to replicate this finding and verify whether app-based CBT-I is effective in preventing both first-onset and recurrent depression. Second, clinical interviews using MINI were administered via telephone instead of face-to-face interviews. However, MINI telephone interview has been widely

adopted in research contexts as a reliable method for assessing psychiatric disorders [44]. Third, although the assessors were masked to group allocation when conducting telephone interviews, it was possible that the assessors may still become aware of participants' allocations based on their responses. Fourth, app-based HE with the inclusion of general sleep knowledge may not be a completely 'inactive' control condition [45], as the controls similarly improved in both insomnia and depression outcomes, albeit to a lesser extent than the intervention group. Last, the modest sample size of adolescents (<18 years, $n = 10$) limits the generalizability of the findings to this age group.

## 5. Conclusions

The current study documented the efficacy of app-based CBT-I for preventing future depression at both symptom and disorder levels, as well as reducing nocturnal and daytime impairments among high-risk youth. These findings support the integration of digital CBT-I into clinical practice for the youth population.

## Supporting information

**S1 Protocol. Effectiveness of e-based cognitive behavioral therapy for insomnia on improving mental health in Chinese youths with insomnia: a large-scale randomized control trial.**
(PDF)

**S1 Checklist. CONSORT 25-item checklist.** CONSORT, Consolidated Standards of Reporting Trials.
(DOCX)

**S1 Appendix. Supplementary appendix.** Table A. Description and completion rates of app-based CBT-I and app-based health education (HE) sessions during the intervention period. Table B. Hazard ratio (HR) of incident major depression in app-based CBT-I group as compared to app-based HE. Table C. HR of incident major depression in app-based CBT-I group as compared to app-based HE among participants with persistent insomnia. Table D. Secondary outcomes with imputed missing data due to participants reaching study endpoint before final follow-up. Table E. Mediation analysis results. Table F. Other prespecified outcomes. Table G. Other prespecified outcomes with imputed missing data due to participants reaching study endpoint before final follow-up. Fig A. Sleep-promoting medication use by intervention group. Fig B. Risk of developing major depressive disorder by subgroups. Fig C. Risk of developing major depressive disorder by subgroups among participants with persistent insomnia. Fig D. Remission rates of insomnia disorder by intervention group with imputed missing data due to participants reaching study endpoint before final follow-up. Fig E. Comparison of secondary outcomes at each assessment. Fig F. Comparison of other prespecified outcomes at each assessment.
(DOCX)

## Acknowledgments

We thank all the partner schools and participants for their cooperation and participation, BestCare & SuMian BioTech Co, Ltd. for their support of this research, and the content creators in social media for promoting our study. The study was carried out by the Li Chiu Kong Family Sleep Assessment Unit, Department of Psychiatry, The Chinese University of Hong Kong and Peking University Sixth Hospital. The insomnia intervention program was provided to all the trial participants at no cost by our research team (Li Chiu Kong Family Sleep

Assessment Unit, Department of Psychiatry, The Chinese University of Hong Kong, Hong Kong SAR) and BestCare & SuMian BioTech Co, Ltd.

## Author contributions

**Conceptualization:** Ji-Hui Zhang, Lin Lu, Yun Kwok Wing.

**Formal analysis:** Si-Jing Chen, Ji-Hui Zhang.

**Funding acquisition:** Lin Lu, Yun Kwok Wing.

**Investigation:** Si-Jing Chen, Jian-Yu Que, Ngan Yin Chan, Le Shi, Shirley Xin Li, Joey Wing Yan Chan, Weizhen Huang, Chris Xie Chen, Chi Ching Tsang, Yuen Lam Ho, Charles M. Morin, Ji-Hui Zhang, Lin Lu, Yun Kwok Wing.

**Methodology:** Si-Jing Chen, Jian-Yu Que, Ngan Yin Chan, Le Shi, Shirley Xin Li, Joey Wing Yan Chan, Weizhen Huang, Chris Xie Chen, Chi Ching Tsang, Yuen Lam Ho, Charles M. Morin, Ji-Hui Zhang, Lin Lu, Yun Kwok Wing.

**Project administration:** Si-Jing Chen, Jian-Yu Que, Chi Ching Tsang.

**Supervision:** Ji-Hui Zhang, Lin Lu, Yun Kwok Wing.

**Writing – original draft:** Si-Jing Chen, Ji-Hui Zhang, Yun Kwok Wing.

**Writing – review & editing:** Si-Jing Chen, Jian-Yu Que, Ngan Yin Chan, Le Shi, Shirley Xin Li, Joey Wing Yan Chan, Weizhen Huang, Chris Xie Chen, Chi Ching Tsang, Yuen Lam Ho, Charles M. Morin, Ji-Hui Zhang, Lin Lu, Yun Kwok Wing.

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
