## [Editor Report · Decision Letter 0]

23 Aug 2024

Dear Dr Wing,

Thank you for submitting your manuscript entitled "Effectiveness of E-based Cognitive Behavioral Therapy for Insomnia on Preventing Major Depressive Disorder in Youth with Insomnia and Subclinical Depression: A Randomized Clinical Trial" for consideration by PLOS Medicine.

Your manuscript has now been evaluated by the PLOS Medicine editorial staff and I am writing to let you know that we would like to send your submission out for external peer review.

Please re-submit your manuscript within two working days, i.e. by Aug 27 2024 11:59PM. Please do let us know if you need more time.

Kind regards,

Syba

Syba Sunny, MBBS, MRes, FRCPath

Associate Editor

PLOS Medicine

ssunny@plos.org

---

## [Decision Letter · Decision Letter 1]

22 Oct 2024

Dear Dr Wing,

Many thanks for submitting your manuscript "Effectiveness of E-based Cognitive Behavioral Therapy for Insomnia on Preventing Major Depressive Disorder in Youth with Insomnia and Subclinical Depression: A Randomized Clinical Trial" (PMEDICINE-D-24-02778R1) to PLOS Medicine. The paper has been reviewed by subject experts and a statistician; their comments are included below and can also be accessed here: [LINK]

As you will see, whilst there was an appreciation of the value of the work presented, the reviewers raised a number of concerns, including discrepancies between what was written in the manuscript and the protocol documentation provided. However, after discussing the paper with the editorial team and an academic editor with relevant expertise, I'm pleased to invite you to revise the paper in response to the reviewers' and editors’ comments. We plan to send the revised paper to some or all of the original reviewers, and we cannot provide any guarantees at this stage regarding publication.

We ask that you submit your revision by Nov 12 2024 11:59PM. However, if this deadline is not feasible, please contact me by email, and we can discuss a suitable alternative.

Don't hesitate to contact me directly with any questions (ssunny@plos.org).

Best regards,

Syba

Syba Sunny, MBBS, MRes, FRCPath

Associate Editor

PLOS Medicine

ssunny@plos.org

Comments from the academic editor:

The academic editor was supportive of inviting you to revise your paper. He agreed with the reviewer comments and asked that you give special attention to the comments of Reviewer 3 and the diagnostic issue.

Comments from the reviewers:

Reviewer #1: The paper reports on a trial of an app-based insomnia intervention to prevent depression in young people with insomnia symptoms. I was not a reviewer of the original version of the manuscript and have not been provided with details of the previous review round, so I apologize for any overlap with previous comments. Overall the trial is clearly reported and the outcomes further establish the importance of reducing insomnia symptoms for the prevention of mental ill health. To my knowledge, no previous study has robustly tested whether treatment of insomnia symptoms can prevent depression in a young adult sample.

1. The term "e-based" is not commonly used in the literature and not clearly defined. Electronic interventions are broader than digital interventions or internet interventions, which include smartphone applications (apps). A more precise term should be used throughout.

2. Page 4: "...emergence of sleep and mental health problem[s]" (should be plural)

3. Page 4: "...translate the modifiable risk factors for depression into [an] effective preventive strategy."

4. Page 5: It appears that insomnia disorder was only assessed using the ISI, with no clinical confirmation. Perhaps the term "insomnia disorder" should be replaced by "significant insomnia symptoms" or "probable insomnia disorder". This also seems inconsistent with the trial registration, which indicates that ICD-10 criteria were to be used to assess insomnia disorder.

5. Page 5: "in addition to any pharmacological treatments" - should this be "irrespective of any pharmacological treatments"?

6. Page 5: "The study was conducted and reported in accordance with the CONSORT Guidelines." - CONSORT guidelines are only for the reporting of trials, not a recipe for conducting trials.

7. Page 5: "Informed consents were obtained... consents from their parents" - consent should be singular (unless there were multiple consents provided for different aspects of the trial - if so, this should be described).

8. Page 6: Were the interventions delivered in Chinese? Was any co-design or user testing conducted before the trial?

9. Page 6 / Table S1: How much sleep hygiene content was included in the control condition? The authors are likely aware that sleep hygiene can be an active intervention (e.g., https://doi.org/10.1093/fampra/cmx122). Perhaps this might explain the (smaller) improvements in insomnia in the control condition.

10. Page 8: It is not clear why the sensitivity analysis (LOCF) used a less rigorous methodology than the primary analysis, as LOCF assumes MCAR instead of MAR.

11. Page 9: "attended the 6-month assessments" - the method suggests that assessments were self-completed (was this online?) but the term "attended" suggests they were done in-person at a specified location - please clarify.

12. No adverse events were reported. This seems surprising - did none of the control condition participants report deterioration of symptoms or any other negative experience that might have been related to the intervention? Perhaps the wording of the question about adverse events set a high threshold - some explanation would be helpful.

13. The discussion (p15) reports on impacts on functional impairments - it seems this is based on daytime fatigue. It is not clear that this measure can appropriately be described as assessing functional impairments.

14. The discussion attributes the low attrition to reminders. However, previous similar trials also used automated reminders (e.g., 11). I wonder if the higher adherence might be partly explained by cultural differences in expectations around completion of trial activities, given this trial seems to be the first of its kind conducted outside of Western nations? More could also be made of this outcome - that the effectiveness of addressing insomnia symptoms seems to extend beyond US/Europe/Australia.

15. It is interesting that the intervention had much more modest effects on anxiety (which is also closely tied to insomnia) than on depression. No interpretation of this finding is provided.

16. The limitation around use of the MINI might be strengthened. The measure had modest psychometric performance in the single validation study that was conducted against clinical interviews (in 1998) and I am not aware of any cultural adaptation of the inventory for Chinese-speaking populations.

17. Another limitation of the study that should be noted is that a clear majority of the participants were young adults (18-25) rather than adolescents. There may be development differences in the role of insomnia in adolescents vs young adults, but the adolescent sample in this trial (n=20) was insufficient to examine this possibility and the findings may not generalize to adolescents (under 18 years of age).

Reviewer #2: This paper reports the main results of a randomised controlled trial of an e-based CBT on preventing MDD in youth with insomnia. Overall, I found the paper very clear and easy to follow. The results were consistent across a range of outcomes which strengthen the likelihood of intervention benefits. Most of my comments are relatively minor and are listed below.

Major / general comments:

1) The outcome hierarchy is not entirely clear to me. According to the protocol, outcomes were classified as follows:

* Primary outcomes (x4): ⁕ Remission rate of insomnia disorder conformed by ICD-10 Classification of Mental and Behavioral Disorders, ⁕ Change of insomnia symptoms measured by ISI ⁕ Occurrence of MDD conformed by MINI and ⁕ Change of depressive symptoms measured by PHQ-9

* Secondary outcomes (x2): ⁕ Incidence of suicidality which includes plans and attempts as measured by MINI and ⁕ Change of anxiety symptoms measured by GAD-7

* Other outcomes (x10): ⁕ Incidence of suicidal ideation measured by BSSI, ⁕ Change of daytime symptoms measured by MFI, ⁕ Change of sleep-related thoughts and behaviours measured by DBAS-16, ⁕ Change of circadian rhythms measured by 7-Day Daily Sleep Diary and MEQ, and ⁕ Change of sleep parameters including time in bed (TIB), total sleep time (TST), sleep onset latency (SOL), wake after sleep onset (WASO), sleep efficiency (SE) measured by 7-Day Daily Sleep Diary

While all these outcomes appear to be reported in the manuscript, time to MDD onset has been selected as the primary outcome with PHQ9 and ISI as secondary outcomes. Please justify this change/choice.

2) The primary outcome is time to MDD onset (i.e. a survival outcome); however, assessments only occurred at discrete visits (6 and 12 months). Based on the discrete timing of the assessment, I would expect censoring and the recording of event times to occur only at the time of visits but the Kaplan-Meier plot suggests that event times were recorded continuously which surprises me. Was the actual date of MDD onset recorded (e.g. using a diary)? Please clarify in the response and manuscript.

3) The attached protocol and SAP are in a Word document which does not contain any contextual information e.g. title page, date or footnote. If available, please share a full version of the protocol (and SAP) e.g. in PDF format.

Minor / specific comments:

4) The sample size calculation assumes a HR of 0.5. This is a fairly large effect, -at least for someone new to this field such as me -; however, there is no justification for the choice of effect size in the manuscript. Please add.

5) "Remission rates of insomnia were analysed using a weighted generalized estimating equations model". It is not clear to me why a "weighted" version of GEEs was used. Please clarify the purpose and weighting approach.

6) When reporting the results of new onset MDD, the authors indicate that 10% and 18% of participants developed new onset in the intervention and control groups respectively. In the same paragraph, they report the one-year incidence of MDD estimated from the cumulative incidence curves. These numbers (12% and 21%) are different from the proportions at one year (10% and 18%). The differences are presumably due to censoring when using cumulative incidence curves. Please confirm/clarify.

7) Adjusted analyses of new onset MDD were conducted after adding multiple covariates (including prior history of MDD, educational level, family income, comorbid medical illnesses and use of sleep-promoting medication). Please clarify whether this adjusted analysis was pre-specified and how the covariates were selected. Were other adjusted analyses performed?

8) In the results section 3.3.1 (remission rates - bottom of page 12), please clarify what is meant by "weighted" in "the *weighted* percentage of remitters". Please also clarify what is mean by "controlling missing data" in "after controlling for strata variables and *missing data*". Does this mean that missing data was imputed? And if so, how? I would expect the primary analysis of remission rates over time to be based on all available data with no imputation; however, it is not clear to me that this was the approach used (except in sensitivity analyses).

9) Table 2. Please add the number of participants with available data at each timepoint for each arm and outcome when reporting the mean and SE by visit.

10) Please consider including plots of PHQ-9 and ISI (and potentially PHQ-8) scores showing the mean and SE by treatment for each visit to supplement Table 2. These plots could be part of the supplementary appendix. The same could be done for other outcomes presented in S5 Table.

-Laurent Billot

Reviewer #3: The authors investigated the effects of digital insomnia cognitive behavioral therapy on the occurrence of major depression in adolescents through a randomized, double-blind clinical trial. As a result of the 12-month follow-up, the prevalence of insomnia and the incidence of major depression in the group receiving digital insomnia cognitive behavioral therapy were significantly lower than those in the control group.

The strength of this study is that it was conducted while controlling various variables well. This study is meaningful in that there is a lack of studies that control variables that affect the results well. Another strength is that the compliance of the study participants was evaluated and presented.

However, it is a significant problem that the duration of insomnia as a study participant was not applied as the diagnostic criterion for insomnia adopted by DSM-IV and ICSD-3 since 2013, "3 months." In addition, previous studies on insomnia and depression have shown that persistent insomnia affects relapse rather than onset of depression (Persistent Sleep Disturbance: A Risk Factor for Recurrent Depression in Community-Dwelling Older Adults, http://dx.doi.org/10.5665/sleep.3128). It is an important issue whether the variables evaluated by the authors measure relapse or first onset of depression. In addition, the authors should have considered the history of depression when randomizing. Third, the authors should mention the mechanism of operation of the digital cognitive behavioral therapy for insomnia. Information should be provided on whether the therapist delivers the prescription through the Internet based on the entered sleep diary or by an artificial intelligence model, what mechanism is used to prescribe the sleep schedule provided to the study participants, and whether cognitive therapy also addresses depression or only insomnia. It is necessary to describe the aforementioned issues as limitations. Finally, the authors have already mentioned the limitations, such as the difficulty in maintaining double-blindness and the fact that the recurrence of depression was assessed over the phone, which are important issues to consider when interpreting the study results.

---

Comments and requests from the editorial team:

* We note that the reviewers picked up on discrepancies between the study protocol that was submitted and the information provided in the main text of the manuscript. For example, it does not appear that the diagnosis of insomnia was made as planned. Please clarify and explain all discrepancies between the paper and protocol. If the outcomes were not prespecified in the protocol, please define them in the Methods (Outcomes section) as post hoc and explain why they were added. Post hoc comparisons should be presented as hypothesis generating rather than conclusive.

* It appears that the study protocol and statistical analysis plan provided may not be the original study documentation. Please also provide original versions of these documents with your next submission.

* Thank you for providing a completed CONSORT checklist. We ask that you kindly revise this and use section and paragraph numbers, rather than page numbers. This is because page numbers tend to change during the production process.

* Please note PLOS’s requirements for data availability (which can be accessed here: https://journals.plos.org/plosmedicine/s/data-availability#loc-faqs-for-data-policy). We ask that you look through this and kindly revise your Data Availability Statement.

Also, note that we cannot accept an author’s email address as the point of contact for data requests. When possible, we recommend authors deposit restricted data to a repository that allows for controlled data access. If this is not possible, directing data requests to a non-author institutional point of contact, such as a data access or ethics committee, helps guarantee long term stability and availability of data. Providing interested researchers with a durable point of contact ensures data will be accessible even if an author changes email addresses, institutions, or becomes unavailable to answer requests.

* Please express the main results with 95% CIs as well as p values. When reporting p values please report as p「0.001 and where higher as the exact p value p=0.002, for example. Throughout, suggest reporting statistical information as follows to improve clarity for the reader "22% (95% CI [13%,28%]; p「/=)". Please be sure to define all numerical values at first use.

FIGURES AND TABLES

SUPPLEMENTARY MATERIAL

RCTs

* Please structure the Methods section using the following sub-headings: Study design and participants, Randomization and masking, Procedures, Outcomes, Statistical analysis.

* Please ensure that all prespecified outcomes (primary, secondary, and exploratory) are listed in the Methods/Outcomes section and indicate whether there are outcomes that are not presented in the current report.

* Please specify the dates (Month Day, Year) during which study enrollment and follow up occurred.

* Please include absolute numbers wherever you report percentages; eg, n/N (%)

* Please present the safety data for the study including numbers of specific events and whether or not adverse events are thought to be related to treatment. AEs should be reported in the abstract, per CONSORT and CONSORT-Harms.

* If your trial had to undergo important modifications in response to extenuating circumstances, please complete the CONSERVE-CONSORT checklist and provide in your Supporting Information; (https://www.equator-network.org/reporting-guidelines/guidelines-for-reporting-trial-protocols-and-completed-trials-modified-due-to-the-covid-19-pandemic-and-other-extenuating-circumstances-the-conserve-2021-statement/). When completing the checklist, please use section and paragraph numbers, rather than page numbers.

* In keeping with our commitment to Open Science, please include the study protocol document and analysis plan (including any amendments) as Supporting Information to be published with the manuscript if accepted.

GENERAL REQUESTS

* Please upload any figures associated with your paper as individual TIF or EPS files with 300dpi resolution at resubmission; please read our figure guidelines for more information on our requirements: http://journals.plos.org/plosmedicine/s/figures. While revising your submission, please upload your figure files to the PACE digital diagnostic tool, https://pacev2.apexcovantage.com/. PACE helps ensure that figures meet PLOS requirements. To use PACE, you must first register as a user. Then, login and navigate to the UPLOAD tab, where you will find detailed instructions on how to use the tool. If you encounter any issues or have any questions when using PACE, please email us at PLOSMedicine@plos.org.

---

## [Decision Letter · Decision Letter 2]

13 Dec 2024

Dear Dr. Wing,

Thank you very much for re-submitting your manuscript "Effectiveness of App-based Cognitive Behavioral Therapy for Insomnia on Preventing Major Depressive Disorder in Youth with Insomnia and Subclinical Depression: A Randomized Clinical Trial" (PMEDICINE-D-24-02778R2) for review by PLOS Medicine.

I have discussed the paper with my colleagues and the academic editor and it was also seen again by the reviewers. I am pleased to say that provided the remaining editorial and production issues are dealt with we are planning to accept the paper for publication in the journal.

[LINK]

We look forward to receiving the revised manuscript by Dec 20 2024 11:59PM.   

Sincerely,

Rebecca Kirk

On behalf of:

Syba Sunny, MBBS, MRes, FRCPath

Senior Editor 

PLOS Medicine

plosmedicine.org

Requests from Editors:

GENERAL EDITORIAL REQEUSTS

* At this stage, we ask that you include a short, non-technical Author Summary of your research to make findings accessible to a wide audience that includes both scientists and non-scientists. The Author Summary should immediately follow the Abstract in your revised manuscript. This text is subject to editorial change and should be distinct from the scientific abstract. Ideally each sub-heading should contain 2-3 single sentence, concise bullet points containing the most salient points from your study. In the final bullet point of ‘What Do These Findings Mean?’ Please include the main limitations of the study in non-technical language.

Please see our author guidelines for more information: https://journals.plos.org/plosmedicine/s/revising-your-manuscript#loc-author-summary. "

* Please confirm that your title complies with to PLOS Medicine's style. Your title must be nondeclarative and not a question. It should begin with main concept if possible. "Effect of" should be used only if causality can be inferred, i.e., for an RCT. Please place the study design ("A randomized controlled trial," "A retrospective study," "A modelling study," etc.) in the subtitle (ie, after a colon).

* Please confirm that your abstract to complies with our requirements, including providing all the information relevant to this study type https://journals.plos.org/plosmedicine/s/submission-guidelines#loc-abstract

* Please ensure that the Introduction ends with a clear description of the study question or hypothesis.

* Please ensure that all abbreviations are defined at first use throughout the text.

FUNDING STATEMENT

* The funding statement should include: specific grant numbers, initials of authors who received each award, URLs to sponsors’ websites. Also, please state whether any sponsors or funders (other than the named authors) played any role in study design, data collection and analysis, the decision to publish, or preparation of the manuscript. If they had no role in the research, include this sentence: “The funders had no role in study design, data collection and analysis, decision to publish, or preparation of the manuscript.”

COMPETING INTERESTS STATEMENT

* All authors must declare their relevant competing interests per the PLOS policy, which can be seen here: https://journals.plos.org/plosmedicine/s/competing-interests For authors with ties to industry, please indicate whether any of the interests has a financial stake in the results of the current study.

ETHICS AND CONSENT

* Please specify whether informed consent was written or oral. Please ensure that the research complies with the PLOS policy in full: https://journals.plos.org/plosmedicine/s/human-subjects-research#loc-patient-privacy-and-informed-consent-for-publication

FIGURES

* Please provide titles and legends for all figures (including those in Supporting Information files).

CLINICAL TRIALS

* The sample size listed in the submitted manuscript and the trial registry differ. Please explain the discrepancy.

Comments from Reviewers:

Reviewer #1: I thank the authors for their comprehensive responses to my comments. I have no additional concerns.

-- Phil Batterham

Reviewer #2: All my previous comments have been adequately adressed. Thank you.

-Laurent Billot

Reviewer #3: The authors have fully explained and revised the manuscript in response to the reviewers' questions and comments.

[LINK]

---

## [Editor Report · Decision Letter 3]

16 Dec 2024

Dear Dr Wing, 

On behalf of my colleagues and the Academic Editor, Mark Tomlinson, I am pleased to inform you that we have agreed to publish your manuscript "Effectiveness of App-based Cognitive Behavioral Therapy for Insomnia on Preventing Major Depressive Disorder in Youth with Insomnia and Subclinical Depression: A Randomized Clinical Trial" (PMEDICINE-D-24-02778R3) in PLOS Medicine.

PRESS

Sincerely, 

Rebecca Kirk

On behalf of:

Syba Sunny, MBBS, MRes, FRCPath 

Senior Editor 

PLOS Medicine